# Effect of Instability and Bodyweight Neuromuscular Training on Dynamic Balance Control in Active Young Adults

**DOI:** 10.3390/ijerph17238879

**Published:** 2020-11-29

**Authors:** Carla Gonçalves, Pedro Bezerra, Filipe Manuel Clemente, Carolina Vila-Chã, Cesar Leão, António Brandão, Jose M. Cancela

**Affiliations:** 1Faculty of Education and Sport Sciences, University of Vigo, Campus A Xunqueira, Pontevedra, 36005 Pontevedra, Spain; chemacc@uvigo.es; 2Escola Superior Desporto e Lazer, Instituto Politécnico de Viana do Castelo, Rua Escola Industrial e Comercial de Nun’Álvares, 4900-347 Viana do Castelo, Portugal; pbezerra@esdl.ipvc.pt (P.B.); filipeclemente@esdl.ipvc.pt (F.M.C.); ces.leao@gmail.com (C.L.); antoniobrandao@esdl.ipvc.pt (A.B.); 3Research Center in Sports Sciences, Health Sciences & Human Development, CIDESD, 5001-801 Vila Real, Portugal; cvilacha@ipg.pt; 4Instituto de Telecomunicações, Delegação da Covilhã, 1049-001 Lisboa, Portugal; 5Instituto Politécnico da Guarda, 6300-559 Guarda, Portugal

**Keywords:** balance control, Y balance test, training, unstable surface, universitarians

## Abstract

The aims of this study were to analyse the effects of unstable and stable bodyweight neuromuscular training on dynamic balance control and to analyse the between-group differences after the training period. Seventy-seven physically active young adults (48 males, 29 females, 19.1 ± 1.1 years, 170.2 ± 9.2 cm, 64.1 ± 10.7 kg) were distributed into an unstable training group (UTG), a stable training group (STG), and a control group (CG). Training was conducted three times a week for nine weeks. Pre-intervention and post-intervention measures included dynamic balance control using a Y Balance Test (YBT), anterior (A), posteromedial (PM), and posterolateral (PL) reach direction. A mixed ANOVA was executed to test the within-subjects factor and the between-subjects factor. Statistically significant differences were found for all YBT measures within groups (*p* = 0.01) and between groups (*p* = 0.01). After the intervention, UTG and STG presented meaningfully improved results in all YBT measures (A: 7%, *p* = 0.01; 4%, *p* = 0.02, PM: 8%, *p* = 0.01; 5%, *p* = 0.01, PL: 8%, *p* = 0.01; 4%, *p* = 0.04, respectively). No statistical changes were found for any of the measures in the CG. After the intervention, significant differences were observed between the UTG and CG for the YBTA and PM (*p* = 0.03; *p* = 0.01). The results suggest that neuromuscular training using an unstable surface had similar effects on dynamic balance control as training using a stable surface. When compared to CG, UTG showed better performance in YBTA and PM.

## 1. Introduction

Postural and balance control comprise a complex motor function that requires interactions between multiple dynamic sensorimotor processes that are fundamental to safely accomplishing any type of movement performed during daily living [1,2]. This function also involves the coordination of movement strategies and muscular synergies to stabilise the centre of pressure (COP), thus minimising displacements of the centre of pressure when one assumes quasi-static positions, performs certain daily movements, or participates in sports [1]. Proper balance control is the basis for the correct execution of various complex technical movements and improvements in athletic performance [3,4,5]. Furthermore, it reduces the risk of body imbalance and falling [6] and minimises injury risks [7,8,9,10].

Static and dynamic balance have been assessed in many types of individuals and contexts [8,11,12]. However, dynamic balance measures seem to be more relevant for healthy people, young adults, and athletes [13]. Previous studies have described dynamic postural stability measures as being more challenging than static balance tests for young adults [13,14]. Researchers have used different instruments to assess dynamic balance. Some of these instruments include the Balance Error Scoring System (BESS) [15,16], the jumping test [17], the Star Excursion Balance Test (SEBT) [15,18,19,20], and the Y Balance Test (YBT) [12,21,22,23,24]. The SEBT was developed to assess lower extremity dynamic balance and stability to detect reach deficits in participants with lower extremity injuries [25,26,27], as well as to predict lower extremity asymmetry and injuries [21,28,29,30]. In an effort to reduce redundancy and improve efficiency, the SEBT was modified to become the YBT, which assesses only the anterior, posteromedial, and posterolateral reach directions [28,31,32]. The dynamic postural control measured by this instrument presents a significant challenge regarding the postural control system, as it requires a functional task to be completed without a loss of balance. The advantage of this method is that it imposes additional demands related to proprioception, range of motion, and strength while the participant remains upright and steady [33].

Previous research has also investigated the influence of training programs on dynamic balance control. Some studies have studied the effects of training programs involving exercises performed on stable and unstable surfaces for the three planes of motion (sagittal, frontal, and transverse) while assuming a unipedal and bipedal stance, with or without recurrent balance challenges (i.e., ball throwing or catching, or external perturbation applied by a partner) [2,15,20,24]. The existing research seems incongruent relative to the real effects of training programs on dynamic balance. Some studies on athletes did not report any significant influence of balance training on dynamic balance [17,20,32]. However, other researchers have reported that balance exercises, resistance or functional exercises, and proprioceptive/neuromuscular training programs effectively improve dynamic balance [12,15,18,22,24,34,35]. For example, Alyson et al. [35], Benis et al. [24], and Mcleod et al. [15] studied the effect of a neuromuscular training program on dynamic balance control in athletes and reported that the training program improved dynamic balance, postural control, lower limb stability, and proprioceptive capabilities [15,24,35].

However, the impact of bodyweight neuromuscular training on athletes’ dynamic balance is not well-understood. For a better prescription of balance exercises among this population, it may be important to research the benefits of applying such training programs and to understand some specific characteristics of their performance. It is essential to know the implications of these training programs in terms of their practical application. Therefore, the aim of this study was two-fold: (i) to analyse the effects (within-group differences) of a nine-week unstable and stable bodyweight neuromuscular training program on dynamic balance in active university students, and (ii) to analyse the between-group differences after the nine-week program in active university students.

## 2. Materials and Methods

### 2.1. Participants

The sample consisted of physically active young adults who were university students in a sport and leisure undergraduate course, at 17–22 years of age. Seventy-seven physically active university students voluntarily participated in this study (Table 1).

The participants were randomly assigned via computer distribution to either a control group (CG) or one of two training groups (UTG—unstable training group; STG—stable training group). The participants completed a medical history questionnaire and the international physical activity questionnaire (IPAQ—short form) so that their physical activity could be measured. All participants performed sports activities (outdoor sports, football, basketball, and others) included in the curriculum of their leisure sports course. The inclusion criteria in the study were (1) training at least three days per week, and (2) absence of acute injuries. The exclusion criteria were (1) previous experience with bodyweight neuromuscular training using unstable platforms or sports that develop balance and proprioceptive skills (e.g., dance, ballet, skating, hockey) and (2) neuromuscular diseases, vestibular disorders, cerebral concussions, chronic lower injuries, or any pathology or health problem that might affect balance and postural control [36,37]. All participants gave written informed consent to participate in this research. The study followed the ethical standard for research conducted in humans as established by the Declaration of Helsinki and was approved by the local ethical committee of the Polytechnic Institute of Viana do Castelo, School of Sport and Leisure with the code number (IPVC-ESDL180801).

### 2.2. Experimental Approach to the Problem

This study followed a randomised design (parallel study). Two intervention groups (UTG—unstable training group; STG—stable training group) and one control group (CG) were organised. The anthropometric measures were assessed in a laboratory before breakfast on a weekday (48 h after the last training/exercise session) between 8:30 a.m. and 10:00 a.m. The experts (all sports science specialists) who measured the outcome measures were blinded for the study intervention and participant recruitment. The participants were instructed to avoid exercising for a minimum of 24 h and consuming alcoholic drinks for a minimum of 48 h before testing. Dynamic balance control was measured using a YBT pre-intervention and post-intervention, and within-group and between-group changes were tested. An initial information session was held during which the study design (including the study aims), the pre-intervention and post-intervention assessments dates, the testing protocol, and the intervention plan were explained. The intervention program occurred three times a week for 9 weeks. All training sessions were supervised and led by a single physical fitness expert with more than ten years of experience.

### 2.3. Training Intervention

The training groups (UTG and STG) we required to complete a 9-week training program that involved three supervised sessions per week (on Tuesdays, Thursdays, and Fridays). Each session lasted about 45 min.

The training warm-up consisted of 5–10 min of submaximal intensity aerobic and mobility exercises. The training protocol consisted of bodyweight neuromuscular exercises—particularly lower extremity strength exercises (i.e., squat, lunge, hip abduction, quick side-push away, skiing moguls, single-leg balance, and lateral front run (2–3 sets of 15 reps), with each week being more complex than the previous one and without impairing participants’ technique or safety (challenging the somatosensory, vestibular, and visual systems). The selected exercises, regressions, and progressions were designed to challenge one or more of the sensory systems integral to maintaining balance (i.e., contact points, visual effects, movement, and external stimulus) [4,38,39].

The unstable training group (UTG) performed a bodyweight neuromuscular training program using unstable platforms that had an inflated dome side and a hard rubber flat side (bosu), and were 25 inches in diameter. The dome was inflated to a firm density and a height of about 8–10 inches from the floor. The stable training group (STG) performed similar bodyweight neuromuscular training on the floor. The CG was asked to maintain their daily routines. If the participants attended over 80% of all sessions, they were included in the study. Both of the training groups (UTG and STG) completed the same session number (24 sessions of training).

### 2.4. Anthropometric Measures

All participants wore light clothing and stood barefoot. Standing height was measured to the nearest 0.1 cm using a portable stadiometer (Seca 217, Hamburg, Germany). Body mass was assessed to the nearest 0.1 kg with a mechanical floor scale (Seca 760, Germany). Eight skinfolds (triceps, subscapular, biceps, suprailiac, abdominal, supraspinal, thigh, and calf) were assessed with a Harpenden calliper (British Indicators, Ltd., London, UK). All anthropometric variables (height, weight, skinfolds, and girths) were measured according to the protocol recommended by the International Society for the Advancement of Kinanthropometry (ISAK) by a single certified expert (ISAK Level 2).

Each participant’s dominant limb length was measured in centimetres from the anterior superior iliac spine to the most distal portion of the medial malleolus with a tape measure. Leg length was used to normalise YBT excursion distances by dividing the average of three maximal reaches by leg length, multiplied by 100 [28,33,40].

### 2.5. Dynamic Postural Control

Each participant completed a YBT modelled according to the methodology described by Plisky et al. [28]. The YBT included the anterior, posteromedial, and posterolateral directions (see Figure 1) with excellent intra-rater reliability (intraclass correlation coefficient (ICC) of 0.85–0.91, with 95% confidence intervals ranging from 0.62–0.96) and interrater reliability (ICC = 0.99–1.00 with 95% confidence intervals ranging from 0.92–1) [28]. YBT was evaluated using a commercially available device (OctoBalance, system Check Your Motion, SKU 1008, Albacete, Spain). Measurements were taken as the participant pushed away as far as possible from the target with their opposite leg. While maintaining a single-leg stance, the participant reached with their free limb in the anterior (YBT A), posteromedial (YBT PM), and posterolateral (YBT PL) directions in relation to their stance foot. The participant returned to the starting position without losing balance [12,28].

Before the dynamic balance assessment, verbal instructions were given, and a demonstration of how the test should be performed was provided. Each participant performed four experimental practice trials (unaccounted tests) for each direction so that participants became comfortable with performing the task [41,42]. After two minutes of rest, each participant performed three test trials in each direction (accounted tests) [12,23]. An average of three maximal reaches were calculated and recorded. Ten seconds of rest were provided between individual reach trials [26,31]. A trial was classified as invalid if the participant removed their hands from their hips, did not return to the start position, or failed to maintain a unilateral stance on the platform [23,26,28,43].

### 2.6. Statistical Analysis

Descriptive statistics included mean, standard deviation, and 95% confidence interval (95% CI) values. The normality of the sample was tested using the Shapiro–Wilk test, and the equality of error variances were assessed using Levene’s test (*p* > 0.05). The dependent variables that were examined were anterior (YBT A), posteromedial (YBT PM), and posterolateral (YBT PL) directions in relation to the stance foot.

A mixed ANOVA was performed to test the within-subjects factor (time: pre- and post) and between-subjects factor (groups: UTG, STG, and CG). Mauchly´s test was used to determine the severity of departures from sphericity. The correction for violations of sphericity was executed using the Huynh-Feldt correction for each condition, both pre-test and post-test (Greenhouse-Geisser > 0.75). If significant interactions were detected by the mixed ANOVA, a Bonferroni post hoc test was used (*p* < 0.05). Cohen’s standardised effect size was calculated for the pairwise comparisons. The magnitudes of differences were defined based on the following thresholds [44]: 0–0.2, trivial; 0.2–0.6, small; 0.6–1.2, moderate; 1.2–2.0, large; >2.0, very large. The statistical procedures were executed in SPSS (version 27, IBM, Armonk, NY, USA) (*p* < 0.05).

## 3. Results

Baseline anthropometric characteristics are presented in Table 1.

Significant differences were found between groups in terms of height. Specifically, the CG participants were taller than the UTG participants. No age, body mass or body mass index differences were found between groups (Table 1). Groups were compared for YBT baseline differences. No significant differences between the groups were found for YBT A (*p* = 0.31), YBT PM (*p* = 0.40), or YBT PL (*p* = 0.18).

After the 9-week intervention plan, statistically significant effects were found within groups (*p* = 0.01) and between groups (interaction time × group) (*p* = 0.01) for all YBT measures.

The outcome measures before and after 9 weeks of bodyweight neuromuscular training for the unstable training group (UTG) and stable training group (STG) for measures (YBT A, PM, PL) are presented in Figure 2.

There was a significant training advantage for all conditions (YBT A, PM, and PL) for both experimental groups (UTG and STG); compared to their baseline scores, the UTG and STG showed statistically significant increases in YBT A (*p* = 0.01, Δ = 3.70 and *p* = 0.02, Δ = 2.24, respectively), YBT PM (*p* = 0.01, Δ = 6.02 and *p* = 0.01, Δ = 3.58, respectively), and YBT PL (*p* = 0.01, Δ = 5.48 and *p* = 0.04, Δ = 2.86, respectively) (Figure 2).

The magnitudes of differences for the UTG were moderate for all measures (YBT A, *d* = −0.69; PM, *d* = −0.87; PL, *d* = −0.89) and for STG were small to moderate (YBT A, *d* = −0.50; PM, *d* = −0.62; PL, *d* = −0.49). In contrast, no statistical changes were found in the CG for any measures (Figure 2).

Post-intervention, statistically significant differences were found between the UTG and CG for the YBT A and PM (*p* = 0.03 and *p* = 0.01, respectively). The magnitudes of differences between groups were moderate for YBT A and PM *(d* = 0.73 and *d* = 0.87, respectively).

## 4. Discussion

The purposes of this study were to analyse the effects of a nine-week unstable and stable bodyweight neuromuscular training program on dynamic balance in active university students and to analyse between-group differences after the nine-week period. Overall, the data revealed no statistically significant differences in YBT measures between groups after nine weeks of bodyweight neuromuscular training.

Previous studies have reported that strength training (consisting of squats, leg extensions, lunges, calf raises, and curl-ups) leads to significant increases in dynamic balance (assessed by SEBT) by improving the strength of the muscles of the lower extremities [18]. Additionally, it seems that a neuromuscular training program (comprising plyometric, functional-strengthening, stability-ball exercises, and bodyweight core exercises progressing from a stable to an unstable position) improved participants’ performance on YBT and SEBT reach directions, postural control, and lower limb stability tests [15,22,24,35]. However, these previous findings were not supported by our results. Our findings show no significant differences in the YBT measures between groups after nine weeks of bodyweight neuromuscular training. However, current data suggest a significant training advantage in all conditions (YBT A, PM, and PL) for both experimental groups (UTG and STG). We found that, based on comparisons with baseline scores, UTG and STG significantly improved YBT A (+3.70 cm, *p* = 0.01; +2.24 cm, *p* = 0.02, respectively), YBT PM (+6.02 cm, *p* = 0.01; +3.58 cm, *p* = 0.01, respectively), and YBT PL (+5.48 cm, *p* = 0.01; +2.86 cm, *p* = 0.04, respectively) measures. Both unstable and stable training programs resulted in significant pre- to post-intervention improvements.

The authors studied whether gluteus medius strength training, proprioception training, and a combination of both affect dynamic postural balance (assessed by SEBT) in healthy young adult college students. Their results revealed no significant difference between groups for the SEBT measures. However, all three training programs exhibited significant pre-test to post-test improvements in SEBT measures in comparison to the control group (CG). Additionally, Benis et al. [24] investigated the effects of an eight-week bodyweight neuromuscular training intervention on YBT performance and postural control in basketball players. The authors also found improvements over baseline scores in posteromedial and posterolateral reach directions and the composite YBT scores for the experimental group [24].

In the present study, we found statistically significant improvements in UTG for the YBT A (*p* = 0.03) and YBT PM (*p* = 0.01) after nine weeks of bodyweight neuromuscular training when compared with the CG. No significant differences were found between the STG and CG. Bodyweight neuromuscular training on an unstable surface (UTG) seemed to improve dynamic balance control and anterior and posteromedial maximal reach to a greater extent than the STG and CG. This finding is consistent with the results of Benis et al. [24] and Filipa et al. [35]. The authors also found differences between the experimental group and CG, as the experimental group demonstrated improvements in posteromedial reach, composite scores [24,35], and in the posterolateral direction for both lower limbs [35]. Perhaps the bodyweight neuromuscular exercises performed on unstable surfaces prescribed in the present training protocol promoted lower limb stability (support leg) and mobility (opposite leg) in the A and PM maximal reach directions.

Another noteworthy finding from the present study was that all groups demonstrated worse values for the YBT A measure in comparison to the PM and PL measures, both at baseline and after the intervention. This finding suggests that the anterior reach requires different skills than posteromedial and posterolateral reaches. A specific process that links the central and peripheric nervous system seems crucial to this process. It also seems as though a different muscular strategy is needed to perform anterior maximal reach. In fact, similarly to the present study, some studies that applied a YBT to assess dynamic balance have shown poor values for YBT anterior reach [21,23,28,29,33,40].

## 5. Conclusions

Bodyweight neuromuscular training using an unstable surface does not improve dynamic balance control to a significantly greater extent than using a stable surface. The unstable and stable training programs both resulted in significant pre- to post-intervention improvements. Overall, after nine weeks of bodyweight neuromuscular training, we found significant training advantages in all conditions (YBT A, PM, and PL) for both experimental groups (UTG and STG). Although there was no significant difference between groups, the unstable training group showed better performance on the YBT A and PM when compared to the CG. The YBT A seems to require different skills than the YBT PM and PL maximal reaches. In the present study, all groups demonstrated their worst performance on the YBT A measure, both at baseline and after the intervention.

### 5.1. Limitations and Recommendations for Future Studies

Some limitations of the present study should be considered. For example, the small sample size reduces the generalisation of the data. Additionally, in the present study, we did not assess lower limb strength, core muscle activation, or joint stability and mobility during each YBT direction reach assessment. We programmed only lower limb exercises; core exercises might also have an impact on balance control. Additionally, both male and female participants underwent the same training intervention in this study. The portion of female participants in our study was UTG (70%), STG (53%), and CG (13%); this unbalanced gender distribution might have affected the results because the relative intensity of the training might be different across genders.

Thus, future studies might include core stability, plyometrics, and upper body exercises. In our study, we assessed the dynamic balance control involved in YBT reach directions with the dominant leg; we recommend that future studies assess dynamic balance control in both legs and verify possible asymmetries. Furthermore, we suggest that samples in future works should be more representative than in the present study. Similarly, the study groups should be more balanced regarding the numbers of female and male participants. Finally, it would be interesting to investigate the effects of the same training program and methodology in beginners and athletes with a returning injury.

## 5.2. Practical Applications and Study Relevance

Both training groups (UTG and STG) completed the study with a greater increase in YBT measures in comparison to the CG. Overall, bodyweight neuromuscular training using unstable and stable surfaces provided similar benefits for dynamic balance measures in physically active young adults. However, the unstable surface seemed to promote greater improvements in most of the YBT reach directions, consequently leading to better dynamic balance control. Training on unstable surfaces instead of stable surfaces might be useful for challenging physically active young adults’ neuromuscular systems and improving dynamic balance control.

Both training methods (particularly when using an unstable surface) can be used by recreational or competitive active individuals to improve their performance by imposing challenges on their balance control and neuromuscular systems.

## Figures and Tables

**Figure 1 ijerph-17-08879-f001:**
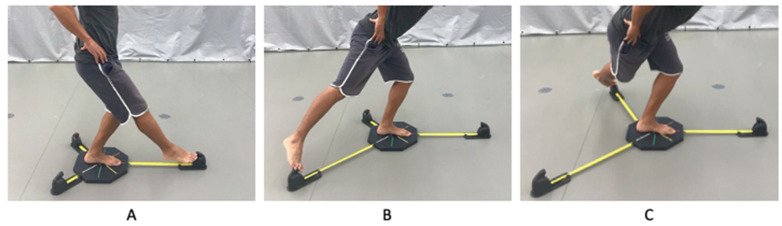
(**A**) Y balance test anterior reach direction; (**B**) Y balance test posteromedial reach direction; (**C**) Y balance test posterolateral reach direction.

**Figure 2 ijerph-17-08879-f002:**
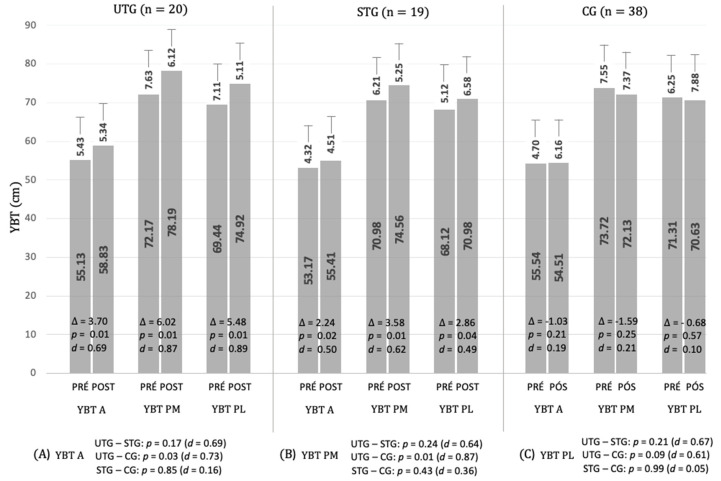
Mean (±SD) of Y Balance Test measures after the 9-week intervention. YBT—Y Balance Test; A—anterior reach direction; PM—posteromedial reach direction; PL—posterolateral reach direction; UTG—unstable training group; STG—stable training group; CG—control group; (Δ) change from baseline to post-intervention; *p* (values) for the difference in pre- and post-measures; *d* = effect size: 0.0–0.2, trivial; 0.2–0.6, small; 0.6–1.2, moderate; 1.2–2.0, large; >2.0 very large. (**A**) Significant differences between groups for YBT A; (**B**) significant differences between groups for YBT PM; (**C**) significant differences between groups for YBT PL.

**Table 1 ijerph-17-08879-t001:** Baseline anthropometric characteristics of the population studied, separated by group (mean ± SD).

Groups	Age (Years)	Height (cm)	Body Mass (kg)	BMI (kg/m^2^)
UTG				
Total (*n* = 20)	19.3 ± 1.1	165.9 ± 9.5	61.3 ± 10.7	22.2 ± 2.5
Males (*n* = 6)	19.5 ± 1.4	176.3 ± 8.3	70.1 ± 13.8	22.4 ± 2.6
Females (*n* = 14)	19.2 ± 1.1	161.4 ± 5.8	57.5 ± 6.5	22.1 ± 2.5
STG				
Total (*n* = 19)	18.8 ± 1.0	169.8 ± 8.1	63.5 ± 10.1	22.0 ± 2.3
Males (*n* = 9)	19.1 ± 1.1	175.8 ± 5.6	70.4 ± 8.9	22.8 ± 2.6
Females (*n* = 10)	18.6 ± 1.0	164.4 ± 6.0	57.3 ± 6.7	21.2 ± 1.9
CG				
Total (*n* = 38)	19.1 ± 1.2	172.7 ± 8.9 *	65.9 ± 10.9	22.0 ± 2.7
Males (*n* = 33)	19.1 ± 1.1	174.8 ± 7.4	67.6 ± 10.4	22.1 ± 2.8
Females (*n* = 5)	19.0 ± 1.4	159.2 ± 5.2	54.5 ± 6.5	21.5 ± 2.5

UTG—unstable training group; STG—stable training group; CG—control group; BIM—body mass index. * Significant differences between sex (*p* < 0.05).

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
