# Peer review of "Effect of Instability and Bodyweight Neuromuscular Training on Dynamic Balance Control in Active Young Adults"

_ijerph, 2020, doi:10.3390/ijerph17238879_

Round 1
Reviewer 1 Report
- I believe that the topic “2.2. Participants” could be moved to 2.1., While “2.1. Experimental approach to the problem” would become 2.2.
- The stretch “physically active young adults who were university students in a sport and leisure undergraduate course” should be in the topic “Participants”. In this same topic it would be important to describe the age of the participants (minimum and maximum)
- In the subtopic “Practical applications and study relevance” it reads: “We prescribed exercises that can be used for the purposes of rehabilitation, injury prevention, and sports training to obtain balance control improvements”. Was there any analysis on the occurrence of sports injuries among the participants? It would be important to comment further on this, although it may be a “study limitation”, especially because the study does not deal with rehabilitation (and even excludes people with injuries).
Author Response
Letter to Reviewer.
The authors want to thank the reviewer for the suggestions for improving the manuscript. The authors recognize that the reviewer suggestions will make the manuscript clearer and more robust.The changes are present in highlight in the manuscript (coloured text).
- I believe that the topic “2.2. Participants” could be moved to 2.1., While “2.1. Experimental approach to the problem” would become 2.2.
Authors: dear reviewer, thank you. The changes were made in the manuscript. The changes are present in highlight in the manuscript (coloured text).
- The stretch “physically active young adults who were university students in a sport and leisure undergraduate course” should be in the topic “Participants”. In this same topic it would be important to describe the age of the participants (minimum and maximum)
Authors: dear reviewer, thank you for your suggestion. The changes were made in the manuscript (coloured text).
- In the subtopic “Practical applications and study relevance” it reads: “We prescribed exercises that can be used for the purposes of rehabilitation, injury prevention, and sports training to obtain balance control improvements”. Was there any analysis on the occurrence of sports injuries among the participants? It would be important to comment further on this, although it may be a “study limitation”, especially because the study does not deal with rehabilitation (and even excludes people with injuries).
Authors: Thank you for your comment. The sentence has been rewritten and references to rehabilitation and injury prevention were deleted, to make it clear (coloured text).
Reviewer 2 Report
The submitted study formulated a hypothesis that has not been fully addressed in previous studies. The way how the authors tested the hypothesis (UTG vs STG) mostly seems technically sound except one major concern.
The total sample size of 77 indicates significant work of the research staffs, but the gender distributions in the three groups are clearly unbalanced; the portion of female participants in UTG, STG and CG are 70%, 53%, and 13%, respectively. According to the manuscript, both male and female participants underwent the same training intervention. Then, it is highly probable that the relative intensity of the training might be different across the two gender groups. Then, this unbalanced gender distribution might affect the result of the submitted study. For example, if the same training intervention imposed higher intensity to female participants, and accordingly affected the female participants more, then the low portion of the female participants in CG might contribute to overestimating the efficacy of the provided intervention. Similar consideration should be applied to the comparison between UTG and STG. Proper justification or discussion regarding this unbalanced gender distribution should be added.
In addition to this major concern, there are some specific points.
1. Figure 2 is poorly presented.
2. Table 2 looks awkward. Type in “Moderate” or “Trivial” in one line. Add lines or anything proper to indicate which of the three YBD measures the information in each line of the last column belongs to.
3. I strongly recommend the authors to receive a professional editing service.
Author Response
The authors want to thank the reviewer for the suggestions for improving the manuscript. The authors recognize that the reviewer suggestions will make the manuscript clearer and more robust.The changes are present in highlight in the manuscript (coloured text).
The total sample size of 77 indicates significant work of the research staffs, but the gender distributions in the three groups are clearly unbalanced; the portion of female participants in UTG, STG and CG are 70%, 53%, and 13%, respectively. According to the manuscript, both male and female participants underwent the same training intervention. Then, it is highly probable that the relative intensity of the training might be different across the two gender groups. Then, this unbalanced gender distribution might affect the result of the submitted study. For example, if the same training intervention imposed higher intensity to female participants, and accordingly affected the female participants more, then the low portion of the female participants in CG might contribute to overestimating the efficacy of the provided intervention. Similar consideration should be applied to the comparison between UTG and STG. Proper justification or discussion regarding this unbalanced gender distribution should be added.
Authors: dear reviewer, thank you. Actually, we do understand your important comment. We have added this potential limitation in the discussion and future research.
In addition to this major concern, there are some specific points.
- Figure 2 is poorly presented.
Authors: dear reviewer, thank you. The figure 2 was improved.
- Table 2 looks awkward. Type in “Moderate” or “Trivial” in one line. Add lines or anything proper to indicate which of the three YBD measures the information in each line of the last column belongs to.
Authors: dear reviewer, thank you. The table 2 was deleted and the outcomes measures of the table 2 are now presented on figure 2, to better understanding.
- I strongly recommend the authors to receive a professional editing service.
Authors: dear reviewer, thank you for your suggestion. Professional editing service feedback has been received.
Reviewer 3 Report
The reviewed manuscript by Goncalves and colleagues compared the effects of a 9-week stable vs. unstable neuromuscular training program in a relatively large cohort of young healthy individuals. The authors observed comparable changes in dynamic balance performance, evaluated using the Y Balance Test, in both groups, and thus concluded that training on an unstable surface does not appear to yield appreciable benefits. Overall, I found the study to be well written and easy to follow and I ask the authors to consider the following minor comments
Abstract
- Please provide descriptive characteristics and gender of study sample
- Please provide data (e.g., changes in balance) as means +/- SE in addition to p-values
- In your summary of findings please specify that this study was in healthy, active young adults
Introduction
- Recommend combining the second and third paragraphs
Methods
- Were there differences in program adherence between groups?
- Did the authors examine the potential of sex as an effect modifier? Certainly, it is possible that men and women would respond differently to the training.
Results
- Something appears to have happened to Figure 2 – the bars are missing and the y-axis is not labeled. Please fix.
- Table 2 - effect sizes should not be negative and I strongly recommend you consider making this into a figure to facilitate easier comparisons among the groups, which are very difficult to interpret in the right-hand column of the table.
Discussion
- The first paragraph of the discussion is unnecessary as this is simply a rehash of information from the introduction. I recommend you start with the second paragraph, which quickly gets to the pertinent information (i.e., overall findings)
- I am uncertain as to the purpose of the third paragraph, which only discusses previous literature without any apparent tie to the findings from the present study
- The authors state that they hypothesized that the unstable surface would provide superior benefits but this was not included in the introduction
Author Response
The authors want to thank the reviewer for the suggestions for improving the manuscript. The authors recognize that the reviewer suggestions will make the manuscript clearer and more robust.The changes are present in highlight in the manuscript (coloured text).
Abstract
- Please provide descriptive characteristics and gender of study sample
- Please provide data (e.g., changes in balance) as means +/- SE in addition to p-values
- In your summary of findings please specify that this study was in healthy, active young adults
Authors: dear reviewer, thank you. The missing information was added to the Abstract.
Introduction
- Recommend combining the second and third paragraphs
Authors: dear reviewer, thank you. The changes were made in the introduction.
Methods
- Were there differences in program adherence between groups?
- Did the authors examine the potential of sex as an effect modifier? Certainly, it is possible that men and women would respond differently to the training.
Authors: dear reviewer, thank you.
- The training groups we required to complete a 9-week training (27 training sessions). Both of training groups (UTG e STG) completed the same session number (24 sessions of training). The missing information was added to the manuscript.
- Actually, we do understand your important comment. We did not examined the potential of sex as an effect modifier, we have added this potential limitation in the discussion and future research.
Results
- Something appears to have happened to Figure 2 – the bars are missing and the y-axis is not labeled. Please fix.
- Table 2 - effect sizes should not be negative and I strongly recommend you consider making this into a figure to facilitate easier comparisons among the groups, which are very difficult to interpret in the right-hand column of the table.
Authors: dear reviewer, thank you for your suggestions.
- The changes were made in the Figure 2
- The table 2 was deleted and the outcomes measures of the table 2 are now presented on figure 2, to better understanding.
Discussion
- The first paragraph of the discussion is unnecessary as this is simply a rehash of information from the introduction. I recommend you start with the second paragraph, which quickly gets to the pertinent information (i.e., overall findings)
- I am uncertain as to the purpose of the third paragraph, which only discusses previous literature without any apparent tie to the findings from the present study
- The authors state that they hypothesized that the unstable surface would provide superior benefits, but this was not included in the introduction
Authors: dear reviewer, thank you.
- The first paragraph has been deleted.
- 3. The paragraph has been rewritten to make it clear.
Round 2
Reviewer 2 Report
Figure 2 still needs to be improved. Most importantly, from the current figure, the readers cannot directly access the comparison among UTG, STG, and CG, which is the most important question that motivated this study (according to the current Introduction). The p values and effect sizes from the statistical analysis that compares each group need to be presented somewhere in the manuscript, preferably in the main figure. Second, fonts are very small to read. In particular, the numbers in the vertical axis are shown in extremely small fonts. All fonts in the figure need to be enlarged. Third, the legend shows that dark gray and light gray indicate mean and SD, respectively, but I cannot find any light grey in the figure. Do the authors mean error bars? Fourth, the ’ mark on the top of “E” of “PRE” needs to be deleted.
In addition, I still find typos; e.g., line 312 “samples n futures” seems to miss “i” in front of “n.” These typos make me highly doubt that the authors received a feedback from professional editors. In the revision, please correct every typo and grammar error, and provide a certificate from the editing service, if any.
Author Response
The authors want to thank the reviewer for the suggestions for improving the manuscript.
Figure 2 still needs to be improved. Most importantly, from the current figure, the readers cannot directly access the comparison among UTG, STG, and CG, which is the most important question that motivated this study (according to the current Introduction). The p values and effect sizes from the statistical analysis that compares each group need to be presented somewhere in the manuscript, preferably in the main figure.
Authors: dear reviewer, thank you. The figure 2 was improved. In fact, we forgot presented the differences between groups after intervention plan. The missing information was added to the figure 2.
Second, fonts are very small to read. In particular, the numbers in the vertical axis are shown in extremely small fonts. All fonts in the figure need to be enlarged.
Third, the legend shows that dark gray and light gray indicate mean and SD, respectively, but I cannot find any light grey in the figure. Do the authors mean error bars? Fourth, the ’ mark on the top of “E” of “PRE” needs to be deleted.
Authors: dear reviewer, thank you. The figure 2 was improved.
In addition, I still find typos; e.g., line 312 “samples n futures” seems to miss “i” in front of “n.” These typos make me highly doubt that the authors received a feedback from professional editors. In the revision, please correct every typo and grammar error, and provide a certificate from the editing service, if any.
Authors: dear reviewer, thank you. In fact, on line 312 an “i” is missing; however, professional editing service feedback were received. I send you the invoice for the editing service.
